Journal of Data-centric Machine Learning Research (2024)          Submitted 10/23; Revised 02/24; Published 03/24

# The Nine Lives of ImageNet: A Sociotechnical Retrospective of a Foundation Dataset and the Limits of Automated Essentialism

**Alexandra Sasha Luccioni**                                      SASHA.LUCCIONI@HUGGINGFACE.CO
*Hugging Face, Canada*

**Kate Crawford**
*University of Southern California,*
*Microsoft Research*

**Reviewed on OpenReview:** *https: // openreview. net/ forum? id= jh0ck1bPGF*

**Editor:** Peter Mattson

## Abstract

ImageNet is on of the most cited and well-known datasets for training image classification models. The people categories of its original version from 2009 have previously been found to be highly problematic (e.g. Crawford and Paglen (2019); Prabhu and Birhane (2020)) and have since been updated to improve their representativity (Yang et al., 2020). In this paper, we examine the past and present versions of the dataset from a variety of quantitative and qualitative angles and note several technical, epistemological and institutional issues, including duplicates, erroneous images, dehumanizing content, and lack of consent. We also discuss the concepts of 'safety' and 'imageability', which were established as criteria for filtering the people categories of the most recent version of ImageNet 21K. We conclude with a discussion of automated essentialism, the fundamental ethical problem that arises when datasets categorize human identity into a set number of discrete categories based on visual characteristics alone. We end with a call upon the ML community to reassess how training datasets that include human subjects are created and used.
*Content warning: This paper contains text and images that some readers may find disturbing, distressing, and/or offensive.*

**Keywords:**  dataset survey, training datasets, ImageNet, image classification, historical analysis, science and technology studies.

## 1 Introduction

Much of the progress and success of modern machine learning (ML) approaches have relied upon creating and disseminating large datasets for training ML models. In computer vision, datasets such as MNIST (LeCun, 1998), ImageNet (Deng et al., 2009) and MS-COCO (Lin et al., 2014) have served as both data for training ML models as well as standardized benchmarks for tracking and comparing the progress of different approaches. In recent years, the pretraining-finetuning paradigm has put even more emphasis on the importance of large datasets, which are necessary for pretraining Transformer-type models (Wolf et al., 2020; Dosovitskiy et al., 2020). It is

therefore crucial to better understand, analyze and document these 'foundation' [1] datasets and what they contain.

ImageNet is one of the best-known and most-used datasets in the ML community; in the last decade, it has become a cornerstone of training ML models and benchmarking progress in tasks such as image classification and object detection. It was first announced in 2009 as generalized computer vision dataset with a novel approach to data collection, based on a subset of categories that were drawn from the WordNet ontology (Miller et al., 1990), then used to query image search engines, and validated by human annotators (Deng et al., 2009). Since then, it has gone through several versions and changes, some of which were instigated by the creators of the dataset, while others proposed by members of the community.

In the current study, we examine ImageNet from a variety of perspectives, ranging from historical (how it evolved over time) and epistemological (according to what theories it was constructed), to ethical (the principles and values underlying how it is made), and sociological (how it is used by the community). We describe the versions of ImageNet have been created over time by both the original dataset creators as well as members of the ML community, and analyze which of these versions have been used for training models. We hone in specifically on the most recent, Winter 2021, version of ImageNet 21K and examine it in terms of the images and categories that it contains and find that despite the changes it underwent, much problematic content both in terms of images as well as entire categories remains. We end with a broader vision of how ML datasets are created and used, and propose steps towards a more principled approach to creating and using datasets that contain images and categories of people.

## 2 Related Work

Since the dissemination of the first ImageNet version in 2010, there have been a multitude of studies of the dataset from different angles, the most relevant of which we describe below:

**Technical Analyses of ImageNet's Quality and its Impacts** Given the size of ImageNet, it is hard to analyze it by hand or via dataset curation methods like embeddings or indexing. However, there have been numerous projects that have analyzed its contents, especially with regards to the extent to which the labels attributed by the labelers correspond to the contents in the images. For instance, a study by Tsipras et al (2020) studied the design choices made during the creation of ImageNet, such as the choice of labels and instructions provided to annotators, and how they impacted the quality of the dataset. Notably, the fact that annotators were instructed to ignore the presence of multiple objects when validating whether a given image corresponded to the label resulted in multiple labels. Another study in the same vein was carried out by Beyer et al., who noted ImageNet's overly restrictive labels overlap and intersect in many cases, and remark a significant number of "semantically and visually indistinguishable groups of images" (2020). They then proposed an improved and re-annotated version of the validation set of ImageNet, which we will describe in more details in Section 3, below. There have also been several studies on the impact that the quality of ImageNet has on downstream model performance (Northcutt et al., 2021; Recht et al., 2019; Stock and Cisse, 2018) and the types of mistakes that ML models make (Vasudevan et al., 2022). All of these analyses have provided useful

---

1. This is a reference to the term 'foundation models' coined by Bommasani et al (2021), who use it to refer to Transformer-type models that can be pretrained once and later fine-tuned for or directly transferred to a multitude of downstream tasks

information for the rest of the ML community about the limitations of the dataset, but also how design choices made over a decade ago continue to shape the ML models and systems of today.

**Sociotechnical Studies of ImageNet's Contents and Methods**   Beyond the more technically-oriented studies above, there have also been important studies that have examined ImageNet from sociotetechnical perspectives. The first such project was carried out by Crawford & Paglen in 2019 (2019), who examined the politics of classification used in ImageNet 21K, with particular attention given to the categories assigned to human beings. This analysis highlighted the series of abusive, outdated and dehumanizing terms in the ImageNet taxonomy – such as "slut", "alcoholic", "mulatto", and "spastic" – which were inherited as part of the WordNet ontology and the archaic values that it encoded. Their study also noted the many non-visual concepts in ImageNet – such as "debtor" and "snob." This work was further continued by Prabhu & Birhane (2020), who found examples of pornographic content as well as underage nudity in the dataset, and reflected on the potential harms that this can have, ranging from perpetuating unjust and harmful stereotypes to providing material for blackmail and voyeurism. Other studies of ImageNet also revealed the geographical and cultural biases it reflects with regards to its portrayal of biodiversity (Luccioni and Rolnick, 2022), and the dynamics of how it is used in practice by the community (Koch et al., 2021; Paullada et al., 2020). Denton et al. (2021) have carried out the most in-depth epistemological commentary of ImageNet to date, which serves as a starting point for the analyses that we carry out in Sections 3 and 5. There is also scholarship on AI datasets in general that has unearthed important insights on the ways datasets are created, used and disposed of in the AI community, which informs our own work (Scheuerman et al., 2021a; Sambasivan et al., 2021; Luccioni et al., 2022).

All of these analyses have helped the ML community reflect upon the fact that ImageNet, despite its popularity and apparent validity from a structural perspective (given the fact that it was based on a hand-curated, formal ontology of concepts), encoded a series of biases that could be transmitted to models trained on this data. They also opened the door towards a more nuanced and complex representation of this massive, highly influential dataset, and the need for a fundamental assessment of the validity of its remediations and methods of human classification. However, in order to better understand ImageNet's present, it is necessary to understand its past, which is why we begin by tracing the history and different versions of ImageNet since it was announced fourteen years ago.

## 3 The Nine Lives of ImageNet

ImageNet remains widely used for a variety of tasks ranging from image recognition to anomaly detection. But what is seen by many as a single, immutable dataset has in fact gone through multiple reincarnations: above and beyond the five official versions that have been made and disseminated by its creators, there have also been at least 4 unofficial versions made by members of the ML community. This means that even on benchmarking leaderboards, there are several versions of the dataset used concurrently, making it difficult to meaningfully compare and replicate results. In the paragraphs below, we endeavor to describe each version of ImageNet, how it was created and how it differs from previous versions, starting with the official versions of ImageNet in Section 3.1 and followed by unofficial versions of the dataset in Section 3.2.

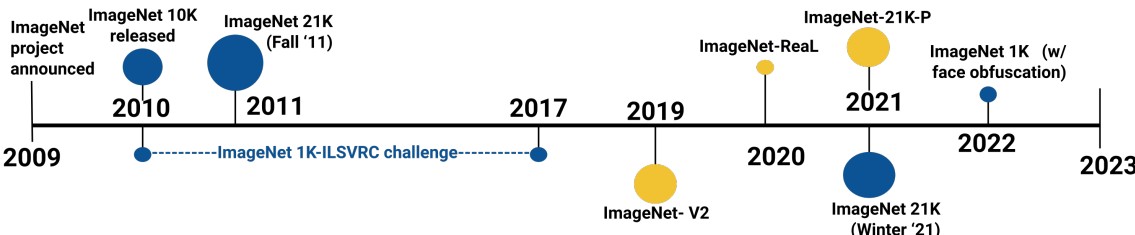

Figure 1: A timeline of of ImageNet versions, spanning from the genesis of the project in 2009 until the present day. Versions in blue are official, whereas those in yellow are those created by the community; the size of each dot is roughly proportional to the size of the version.

## 3.1 Official versions of ImageNet

The ImageNet project was first announced in a 2009 article, which also described its methodology, a novel approach to creating large-scale datasets for machine learning (Deng et al., 2009). The categories, or 'synsets', of ImageNet were picked based on the WordNet ontology (Miller et al., 1990), which also provided a hierarchical backbone to establish relationships between the different categories of labels (e.g. that a 'greyhound' is a type of 'dog', which is a type of 'mammal', which is a type of 'animal', which is a 'living thing'). The dataset was then populated via a two-step process which involved: (1) querying several image search engines [2] using synset names and their translations; and (2) using crowdsourced workers from the Amazon Mechanical Turk to verify the images collected. The article announcing the project stated that, at the time, the dataset was approximately 10% complete, with 3.2 million images from 5,247 categories, and that the aim was to have *"roughly 50 million clean, diverse and full resolution images [made] publicly available and readily accessible online"* (Deng et al., 2009, p.7). Since the initial announcement, several official ImageNet versions were released to the public; their chronology is as follows:

**ImageNet 10K**   The first official version of ImageNet, christened 'ImageNet 10K', was shared in 2010, containing 10,184 categories with more than 200 images each, for a total of 9 million images (Deng et al., 2010). This was the image dataset with the largest number of categories to date, presented as a way of *'reducing the gap between machine performance and human abilities'* (Deng et al., 2010, p.72). The authors tested a variety of ML approaches on the dataset, comparing the training time and performance of different model sizes and architectures. They also describe other, smaller subsets of ImageNet, such as Rand200 – consisting of 200 randomly selected categories – Fungus134 and Vehicle262 – each consisting of descendants of the 'fungus' and 'vehicle' nodes in ImageNet 10K, which are also used for benchmarking purposes in the article but, as far as we could tell, not shared publicly.

**ImageNet 21K (Fall 2011 version)**   The next version of ImageNet was constructed gradually over more than 2 years and eventually grew to contain 21,841 categories (or 'synsets') and a total of 14,197,087 images, and was released to the public in the Fall 2011. It was unofficially coined 'ImageNet 21K' (based on the number of classes) [3], and has been widely used in the last 12 years to train and evaluate several generations of computer vision models. One notable

---

2. The exact search engines used were not specified.

3. Although some also use the term 'ImageNet 22K' since the number of categories is closer to 22,000 than 21,000.

characteristic of the dataset is that, unlike many ML datasets, it does not contain a pre-defined validation split that was shared as part of the dataset. In practice, this meant that model creators would randomly select a subset of the dataset to evaluate model performance.

**ImageNet 1K-ILSVRC (2010-2017 versions)**    In 2010, a subset of the dataset containing 1000 categories was created to be used in the ImageNet Large Scale Visual Recognition Challenge (ILSVRC), a benchmark in image classification and object detection tasks. This subset was used to track the meteoric progress made in supervised computer vision tasks in the 2010s. While in 2011, the top-5 error rate of leading image classification algorithms was around 25%, in 2012 a model called AlexNet (Krizhevsky et al., 2017) achieved an error rate of only 16%, a significant breakthrough at the time [4]. The dataset went through several minor changes in terms of the categories it contained until the 2014 version of the challenge, and has not been updated since. It includes a hidden test set that is hosted on the ImageNet servers, allowing model creators to evaluate their models by uploading a trained model without having access to the labels themselves.

**ImageNet 21K (Winter 2021 version)**    Following academic criticism and public debate (such as the work described in Section 2), further work was carried out on the ImageNet 21K dataset. A new version of the dataset was created by removing those synsets from the 'people' subtree of the dataset that were considered offensive and/or insufficiently visual. The rationale behind this approach was to reduce the "problematic behavior in downstream computer vision technology" (Yang et al., 2020, p.1), namely potentially offensive predictions made by AI models and performance disparities towards underrepresented groups (Crawford and Paglen, 2019; Stock and Cisse, 2018). An updated version of ImageNet 21K, released in 2021, contained 19,167 categories and 13,153,500 images. The filtering was done via a similar crowd-sourcing approach as the creation of the original dataset, which filtered down the 2,832 people categories from the original dataset to only 158. It is currently the only official version of the full ImageNet dataset hosted on the ImageNet website.

**ImageNet 1K-Face Obfuscation (2022)**    The most recent iteration on the 1K subset of ImageNet did not change its categories, but provided an additional set of labels with regards to the presence of human faces in the images themselves as well as a privacy-preserving version of the dataset (Yang et al., 2022b). This intended to address the significant privacy concerns raised about images of people in ImageNet, since the people whose faces appear in the dataset did not provide their consent. While ImageNet 1K only has 3 people categories (the somewhat inscrutable trio of scuba diver, bridegroom, and baseball player), the analysis found that 17% of the images in the dataset contained at least one face. This number was calculated based on a similar two-step procedure as the original ImageNet, whereby a commercial face detection algorithm,Amazon Rekognition, was run on all of the images in the ILSVRC dataset, then the flagged images were validated by human annotators. The study found 562,626 faces in the 1,431,093 images, sometimes with multiple faces per image. The authors also constructed a version of ILSVRC with face obfuscation and showed that using this version for evaluating object recognition approaches resulted in only marginal a loss of accuracy. This version is shared on the ImageNet website alongside the previous versions of ImageNet 1K.

---

4. As of 2023, the error rate stands at less than 9% (Chen et al., 2023)

### 3.2 Unofficial versions of ImageNet

Apart from the official ImageNet versions described above, there are also several versions of the dataset created by members of the community in attempts to address some of the shortcomings that they identified in the official versions, e.g. in terms of labeling schema, image distribution, or efficiency; we describe these below.

**Tiny ImageNet (2015)**  A smaller subset of ImageNet ILSVRC was created in 2015 as part of a Stanford course on Deep Learning for Computer Vision (Le and Yang, 2015). It consisted of 100,000 images from 200 classes, downsized to 64×64 pixels, as well as a test set of 10,000 images. It was dubbed 'Tiny ImageNet' and has since been used as the training and evaluation dataset for the MicroImageNet classification challenge, which runs on Kaggle and continues to be used as a reference for smaller, less computationally-intensive methods, and for trying out the improvement that new approaches for hyperparameter tuning and data augmentation can bring without having to train and test on the larger version of the dataset.

**ImageNet-V2 (2019)**  Given the issue that we noted above regarding a lack of a standardized validation set for ImageNet 21K, an unofficial version was proposed in 2019 by Recht et al (2019), who followed the original process established by the creators of ImageNet to collect 10,000 images to be used for benchmarking image classification models. They also followed the same labeling protocol as the original ImageNet 21K version (providing the same instructions to crowdsourcing workers) and only considered images from a similar time frame as the 2011 version to avoid temporal discontinuities. The authors also evaluated several state-of-the-art models with high reported accuracy on ImageNet 21K and found that they had a lesser performance on their new, 'ImageNet-V2' version, illustrating these models' propensity to overfit to training data and fail to generalize to slightly more difficult images than those found in ImageNet 21K. They then shared their new dataset and called for the community to start using it for a more systematic and generalizable way of comparing image recognition models.

**ImageNet-ReaL (2020)**  In their 2020 paper, Beyer et al (2020), found that many of the gains in accuracy reported by the community on ImageNet ILSVRC are a result of the specificities of the dataset, not improvement of the methods as such. As part of their work, they also collected a new set of labels for the validation set of ImageNet ILSVRC following a "more robust procedure for collecting human annotations" (Beyer et al., 2020, p.1), which includes multiple labels, more accurate labels of objects in images and 3500 images that contain no label at all. Based on these annotations, the authors created a new version of the dataset, which has been named the 'ImageNet-ReaL' benchmark by the community. In the last few years, it is used concurrently with the original ImageNet ILSVRC dataset by many authors in leaderboards and papers.

**ImageNet-21K-P (2021)**  In an effort to make ImageNet 21K more accessible and more efficient for pretraining large neural networks, Ridnik et al (2021a) removed classes with less than 500 labels, reducing the total number of classes by half (to 11,221) but retaining 87% of the images. They also resized all the images to 224x224 pixels and created a standardized validation split in order to facilitate benchmarking and model comparisons. They named the dataset 'ImageNet-21K-P' ('P' for processed) and used it for training a series of models such as Vision-Transformer (ViT) (Dosovitskiy et al., 2020), obtaining equal or better performance to models pretrained on the original ImageNet 21K. The authors also share their code for creating their dataset and it has been used since for training a variety of model architectures (which

we describe in Section 4), given that it improves the speed and efficiency of data processing compared to the original ImageNet version. ImageNet 21K-P is also available for download from the ImageNet website.

What can be gleaned from the timeline and versions of ImageNet described above is that what is colloquially referred to as "the ImageNet dataset" and is used to track progress in computer vision is not, in fact, an immutable dataset, but a multiplicity of sets of images and classes which serve different purposes and transmit an evolving set of values. Members of the ML community have also appropriated the datasets and contributed to it, proposing their own versions that aim to improve its reproducibility and efficiency. How the differing (official and unofficial) versions of ImageNet are used in practice has yet to be established. In the following section, we examine which subsets are the most popular for training computer vision models.

## 4 What versions of ImageNet are being used for training models?

Given the different ImageNet versions described in Section 3, it is unsurprising that many versions of the dataset exist in the wild and are used concurrently for many computer vision tasks. For instance, the popular TensorFlow Datasets library contains 25 different ImageNet versions, most of which are ill-defined versions such as 'few shot ImageNet' and 'corrupted ImageNet'. In terms of benchmarking, the Papers With Code ImageNet Leaderboard presents image classification results on all ImageNet subsets and versions concurrently (providing a series of overlapping tags), making it difficult to consistently track and reproduce results.

In order to get a better understanding of which versions are used by members of the ML community, we downloaded all of the models from the PyTorch Image Models (`timm`) library (Wightman et al., 2019), the largest dedicated repository of pretrained computer vision models [5], that were trained on ImageNet and examined which dataset version was used for training each model. We surveyed a total of 65 models from 14 types of architectures such as MobileNet (Howard et al., 2019), Vision Transformer (Dosovitskiy et al., 2020) and FocalNets (Yang et al., 2022a), and present our findings in Table 1.

Of the 65 models that we examined, we found that only a single model, MViTv2 (Li et al., 2022) used the most recent, Winter 2021, version of ImageNet 21K, even though 20 of the models we examined were trained since its release. In fact, 24 of the models used the Fall 2011 version of the dataset, which is no longer available via the official ImageNet website – which begs the question of how model creators are downloading and sharing it [6]. A further two models use the ImageNet-21K-P version that we described in Section 3. However, it turns out that the majority of the models uploaded on `timm` – 38 in total – are trained on unofficial versions of the dataset developed internally by different organizations.

This includes a Google-created version of the Fall 2011 version of ImageNet 21K, which contains an additional two synsets: 'picture' and 'bleacher', neither of which are in the original dataset – this version was used for training 32 of the 65 models that we surveyed. Microsoft also has their own version of the dataset, with a different label order than that of the official Fall 2011 version, but still containing the problematic categories that were removed in the Winter 2021 version. This version of ImageNet was used to train the FocalNet architecture released in

---

5. The `timm` library has recently been integrated into the Huggingface website, which is how we downloaded the models analyzed.
6. There are multiple versions of ImageNet available on sites such as Academic Torrents and Open Data Lab.

| Model Type | Number of models | Model Year | ImageNet Version | # of classes |
|---|---|---|---|---|
| MobileNetV3 (Howard et al., 2019) | 1 | 2019 | 21K-P | 11221 |
| ConvNeXt (Liu et al., 2022b) | 5 | 2020 | 21K (Fall-2011) | 21841 |
| ResNet-V2-BiT (Kolesnikov et al., 2020) | 6 | 2020 | 21K (Google) | 21843 |
| TResNet (Ridnik et al., 2021b) | 1 | 2020 | 21K-P | 11221 |
| ViT (Dosovitskiy et al., 2020) | 14 | 2020 | 21K (Google) | 21843 |
| BEiT (Bao et al., 2021) | 4 | 2021 | 21K (Fall-2011) | 21841 |
| EfficientNetV2 (Tan and Le, 2021) | 5 | 2021 | 21K (Google) | 21843 |
| MLP-Mixer (Tolstikhin et al., 2021) | 3 | 2021 | 21K (Google) | 21843 |
| Swin Transformer (Liu et al., 2021) | 6 | 2021 | 21K (Fall-2011) | 21841 |
| FlexiViT (Beyer et al., 2022) | 4 | 2022 | 21K (Google) | 21843 |
| FocalNet (Yang et al., 2022a) | 6 | 2022 | 21K (MSR) | 21841 |
| MViTv2 (Li et al., 2022) | 1 | 2022 | 21K (Winter-2021) | 19167 |
| Swin Transformer V2 (Liu et al., 2022a) | 2 | 2022 | 21K (Fall-2011) | 21841 |
| EVA02 (Fang et al., 2023) | 7 | 2023 | 21K (Fall-2011) | 21841 |

Table 1: An analysis of the different versions of ImageNet used for pretraining computer vision models from the `timm` library (Wightman et al., 2019)

2022 (Yang et al., 2022a). More broadly, what we observed is that there is a lack of consistency in the ImageNet versions used for training models that are downloaded tens of thousands of times, making any kind of comparison between these models unreliable. Furthermore, given that these multiple versions of ImageNet exist, there is the possibility that there is an inherent bias in the evaluations conducted by each organization. which uses its own evaluation set. From a sociological perspective, we can note that the remediated ImageNet 21K dataset has not been successfully adopted by the community, since many researchers continue to use previous versions of the dataset, which include the images and categories flagged by critical scholars (Crawford and Paglen, 2019; Prabhu and Birhane, 2020) and the problematic person categories that were identified and ultimately removed by the dataset creators. In the following section, we take a closer look at how this recent version of the dataset differs from the original one, what changes were made, and to what extent they changed the contents of the dataset.

## 5 A Deep Dive into ImageNet 21K (Winter 2021 version)

As described in Section 3.1, the most recent full version of ImageNet was released in winter of 2021 and is currently the only official version disseminated via the project website. Given the critiques made of the original ImageNet 21K dataset, the creators of the updated dataset endeavored to remove problematic categories and images in the new version. In the current section, we reflect upon the ways in which this intention succeeded and where it fell short, supported by empirical evidence gathered by analyzing the dataset.

## 5.1 What Was Removed

As we show in Table 2, the main changes made to ImageNet 21K leading up to its Winter 2021 release involved removing people categories – reducing the original 21,841 categories to 19,167 – as well as images of people – going from 14,197,087 images to 13,153,500.

|  | # of images | # of unique images | # of categories | # of people categories |
|---|---|---|---|---|
| ImageNet 21K (Fall 2011) | 14,197,087 | 12,845,700 | 21,841 | 2,832 |
| ImageNet 21K (Winter 2021) | 13,153,500 | 11,861,470 | 19,167 | 158 |

Table 2: Image and category counts for ImageNet 21K - Fall 2011 and Winter 2021 versions

**Removed Categories**    From the 2,832 people categories that were in the Fall 2011 version of ImageNet, 1,593 of them were deemed potentially offensive and a further 1,081 were considered insufficiently visual, so the 2,674 people categories were reduced to 158 (as reflected in Table 2). Removing many of the particularly problematic people-related categories (such as 'ball-buster', 'bastard' and 'redneck') is an important step towards improving the scope and quality of the dataset, and avoiding that the inherent biases in these categories get perpetuated to downstream models. Due to the fact that the object recognition task is an inherently visual task, categories that were deemed insufficiently visual, such as 'epileptic', 'amoralist' and 'monolingual', were also removed from the dataset. This contributes towards improving the quality of categories that models are trained on, since any information necessary for the task should be contained in the images themselves, not in contextual information about their provenance or characteristics of the individuals depicted in them.

**Removed Images**    Over a million images of people were removed in the Winter 2021 version of ImageNet, belonging to the "problematic" categories described above – i.e. the categories were judged either insufficiently visual or problematic. Many of these images also contain information regarding specific events and the names of photographers that took them, as well as scenes of nudity and violence. As noted in Section 4, given that many of the existing ML models are trained on legacy and custom versions of ImageNet 21K, this means that many problematic images continue to be used in the training of computer vision models.

## 5.2 Remaining Issues in the Dataset

While we recognize that an important curation effort was carried out to create the Winter 2021 version of ImageNet 21K, there are many remaining issues in terms of both the categories and images it contains, which we describe below.

**Category definitions**    We find that the 158 people categories that remain in ImageNet 21K also have other shortcomings, such as categories that conceptually overlap, such as 'professor' and 'assistant professor', or 'tennis player' and 'tennis pro', as well as relative labels such as 'friend' and 'stranger', which can only be defined based on a knowledge of the relationships between different individuals. In both cases, there are definitely people who fall in both categories (i.e. most professors are also friends). There are also time-dependent and region-specific categories, such as 'president' and 'ex-president' (indeed, all of the presidents depicted in

ImageNet are US presidents such as George W. Bush, who have since become ex-presidents). There are also highly stereotypical categories such as 'rapper', which contains only images of Black men, and 'legal representative', which only contains images of men in business suits.

**Duplicates**   Despite the significantly reduced number of people categories, there are still multiple sets of duplicates, such as two different definitions of 'captain' ("the pilot in charge of an airship" and "an officer holding a rank below a major but above a lieutenant") and 'runner' ("someone who travels on foot by running" and "a trained athlete who competes in foot races"), with different sets of images in the two duplicate categories. These categories can be distinguished based on their synset identification numbers (extracted from WordNet), but including duplicate labels has been shown to negatively impact model performance, especially in cases where the two classes are visually undistinguishable (Beyer et al., 2020).

**Erroneous images**   Upon a more in-depth exploration of the dataset, we found that while many people categories have been removed, many of the images contained in those categories still remain in the dataset – in fact, 25,593 images across 1,424 labels from the Fall 2011 version of ImageNet are still included in the Winter 2021 edition as part of a different category, given that many images in the ImageNet 21K dataset were assigned multiple labels [7]. These range from the innocuous images of 'songstress' being re-labeled as 'singer', to the more worrisome 'cover girl' now labeled as 'G-string' and 'lover' as 'bosom'. There is also a number of erroneous images, such as images of non-human animals labeled with people categories like 'sleeper' (shown in Figure 2) and 'zookeeper'.

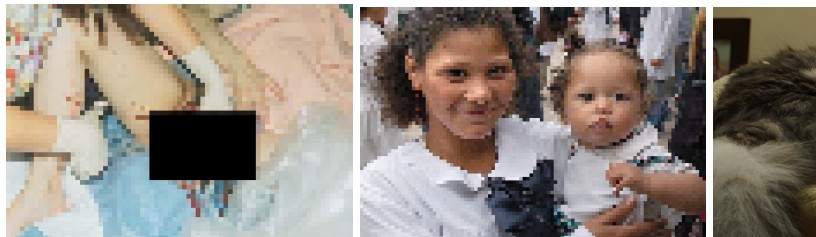

Figure 2: Examples of problematic images of people that remain in the Fall 2021 version of ImageNet, from the labels 'birth' (left), 'schoolchild' (center) and 'sleeper' (right). We have added the black rectangle on the leftmost image, which shows the vulva of a woman giving birth and added pixelation to ensure anonymity.

**Graphic and identifiable images**   Thousands of images of wholly identifiable people, including minors, persist in ImageNet 21K (for instance the center image in Figure 2). There are also multiple graphic images showing scenes of nudity, such as a zoomed-in image of a woman giving birth (leftmost image in Figure 2). The images from both of these categories are, by default, non-consensual, since no consent was ever gathered for the creation or curation of ImageNet.

## 6 Broader Critique and Epistemological Problems

The remediation of ImageNet 21K relies on two key principles of epistemic logic: *safety* and *imageability*. To assess the safety of the dataset, the ImageNet team asked 12 graduate students

---

7. We were able to make these analyses using SeeSet, a dataset exploration tool that we have made for this purpose and will release shortly.

to assess whether an image is "offensive" or "sensitive" - and to flag anything in these categories as "unsafe." The aim was to have, in their words, "100% precision in our labeling of "safe" synsets, meaning that the final list should absolutely have no sensitive or offensive synsets in it." (Yang et al., 2020). This resulted in a dramatic reduction of people categories: from 2,832 to just 158. But even with fewer than 5% of these categories remaining, the images still raise serious concerns from the perspective of what the labels say and what is represented. For example, of the 158 categories of people that are deemed safe, there are categories such as "model", consisting almost entirely of stereotypical images of sexualized women, many of which are pornographic; or "sleeper" which includes many candid images of people asleep on trains, in parks, and in bed, unaware that their photo is being taken; or " birth", which includes graphic images of babies covered in blood. Since the concept of safety is never defined in the paper itself, it is hard to establish for whom these image are meant to be considered "safe". This brings us back to the issue of consent mentioned in Section 5.2 – would the adults or children depicted in these images, be it prone, naked, unconscious, or on a surgeon's table, consent to being included in such a popular dataset?

Furthermore, imageability (defined as "the ease with which the word arouses imagery" (Yang et al., 2020, p.4)) is another core pillar of the recent version of ImageNet 21K. We find that its definition and operationalization represents another epistemic collapse, since it relies upon essentializing human identity in ways that are both scientifically and ethically problematic. This is most obvious in the labeling of gender and race: decades of nuanced scholarship has recognized that human identity is something fluid and culturally constructed, and that forcing people into restrictive categories produces forms of harm, violence, and oppression (Butler, 1990; Bowker and Star, 2000; Thompson, 2015). Yet ImageNet is premised on the assumption that a labeler can simply assign identities including gender and familial relations from an image (see categories like "daughter", "father", "sister", "grandma", "mother" and "mother's daughter"). But the core act of assigning an identity to a person is problematic here: why is a person hanging up their laundry a "washerwoman," or a woman holding a child a "mother", or a person playing a piano a "rockstar"? Of course, many of these categories are contextual – indeed, seeing someone on a bus at 8 am on a Tuesday may indicate that they are a "commuter" – but these labels pertain to people and not surroundings, which are in a different branch of the WordNet taxonomy.

Simply put, it is a category error to imagine that by looking at an image you can define the essence of the person represented: it is *automated essentialism* (Scheuerman et al., 2021b). Labels such as "stranger", "junior", "grandma," or "acquaintance" are fundamentally about a relation - one that is not capturable in an image. A stranger to whom? Junior with regards to what? The labels that have already been determined as imageable show the deep instability of the concept, which has been adopted from work in psycholinguistics – a field which is quite distinct from computer vision. This is a case of using a concept from a different field that does not map to the task, but is used purely because it offers a form of quantification of inherently subjective or non-visual concepts. The attempt at remediating ImageNet not only highlights the limitations of constructing ideas of safety or imageability as inherently quantifiable and automatable on the basis of decontextualized images, but reveals that the core practice of labeling human identity based on an image is inherently problematic – an act of exerting power over a human subject. In the words of Keyes et al., these systems do not find the truth of identity but "naturalize a particular view of it — one that, unsurprisingly, conforms with status quo assumptions." (Keyes et al., 2021, p.165).

# 7 Conclusion

Through our close reading of the many lives of ImageNet in the fourteen years since its initial announcement, we find that despite the numerous attempts of remediation there are several major problems that have not been addressed, ranging from consent to the impossibility of meaningfully evaluating concepts like safety and imageability. Beyond any past or future remediation attempts, there is an unavoidable central philosophical problem for ImageNet and datasets like it: that automated identity detection is fundamentally an essentialist project. It assumes the existence of fixed, essential, and visible attributes that can be used by a central authority to define a human's identity. By attempting to reduce people into fixed and singular categories, this approach ignores the profound and complex interplay between individual agency, social context, and personal experiences. This kind of reductionism both misconstrues the multiplicity of human beings and deprives them of their agency and dignity. This also points to a core lesson for our field: that the practice of automated essentialism, the non-consensual labeling of human identity based on images, is inherently problematic. It deprives people of agency, dignity, and the right of self-determination.

Our findings have several clear implications and paths forward for research for the field of ML. First, ML researchers urgently need substantive ethical guidelines about how images are curated and labeled in training datasets, and in particular, clear ethical guidelines about how inferences are being made about people based on photographs. This could be done via institutionally-based ethical review practices, such as an IRB (Institutional Review Board) specifically geared towards datasets, as well as through ethical guidelines for conferences, such as the NeurIPS Code of Ethics. Initiatives such as the Casual Conversations dataset (Porgali et al., 2023) are also first steps towards consent-driven dataset creation, containing only images of individuals who agreed to participate in the project, were paid for their time, and provided both their age and gender labels themselves. More generally, there is a need for a deeper accounting of, and accountability for, the technical and ethical questions raised by the practice of classifying humans, and the ML community needs to bring more context sensitivity and awareness to ML research and practice.

Furthermore, we need to know more about how ML training dataset, and the models based on them, are propagating throughout academia and industry, and how they are shaping everything from scientific papers to large-scale commercial systems. This is particularly important when ML systems are being used in sensitive contexts, such as medicine, justice and education. There are a number of ML-based approaches for analyzing training datasets to detect specific types of harmful content (such as the ones employed by studies such as Birhane et al. (2023) and Luccioni and Viviano (2021)), which are a good place to start to get a better understanding of the content that is included in both text and image datasets. Finally, given our findings regarding the fact that the most recent version of ImageNet 21K has not been widely adopted in the community, we also believe that researchers should more carefully check dataset versions before using the dataset for training or evaluation, and that the ML community as a whole should adopt clearer deprecation practices for communicating and managing dataset versions (as proposed by Luccioni et al. (2022)), potentially working with data hosting platforms such as Hugging Face (Wolf et al., 2020; Lhoest et al., 2021) to clearly indicate which versions of ImageNet are being hosted and used.

In our current study, we have analyzed which versions of ImageNet are used for pretraining open-source machine learning models and to what extent these versions diverge. But a limitation of our study is that we cannot know how many commercial systems have used it, or how far those

classifications and the images they contain have spread. This becomes a critical problem when datasets are widely used, yet the technical and epistemic problems are only discovered later. This makes it almost impossible to track where the problems are manifesting in commercial systems and where harms are likely to be occurring. As the case of ImageNet and other recent dataset controversies show, it is clear that the appropriation of people's personal media and using it to make sensitive inferences raises deep and complex social issues and is itself a form of representational harm (Barocas et al., 2017). It deserves a more rigorous and careful treatment in both research universities and the industry at large. By conducting retrospective studies such as this one, we can reflect on where lessons have been learned, and where improvements are needed.

## Broader Impact Statement

The ImageNet 21k dataset remains one of the most widely-used datasets in our community; despite this, both its initial creation and its subsequent versions are not well understood. In the current article, we have endeavored to both shed light on the dataset and its evolution, as well as to show how different versions are used in research and practice. We recognize that the examples of problematic people-related categories that we provided can be seen as unacceptable language by many communities, and we strove to be as respectful of that as possible. We have also pixelized the example images provided to ensure that anonymity is respected. Our hope is that our results can shape both the way in which ImageNet is currently used (e.g. encouraging people to use the most recent, official version) as well as the way in which datasets are created in the future (i.e. without assigning categories to human beings). We recognize that creating datasets is a difficult endeavor and we appreciate the curation efforts made by both the original authors as well as members of our community.

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
