# OpenReview forum: "The Nine Lives of ImageNet: A Sociotechnical Retrospective of a Foundation Dataset and the Limits of Automated Essentialism"
_DMLR — Accepted by DMLR_

### Review · Reviewer_Q3xP · 2023-12-22

**Recommendation:** 3
**Confidence:** 2

**Summary Of Contributions:**

This paper mainly discusses the issues found in the people categories of the ImageNet dataset. This paper first examines the people categories through different versions of ImageNet datasets (i.e., official and unofficial ones, old ones and new ones). The authors also take a deep dive in the ImageNet 21k and find there exists ethnical, epistemological and institutional issues in those categories. They also propose the concepts of safety and imageability for filtering the people categories in ImageNet 21k as remediation. In the end, the authors  call the machine learning community to pay attention to the creation of training datasets that include human subjects.

**Strengths:**

1. This paper has a detailed description of the ImageNet dataset from quantitative and qualitative angles and provide a comprehensive standpoint for readers to notice the significance of the issues found in the people categories.
2. The focus on the people categories is reasonable and really important. This is a good perspective for understanding the weakness of current ImageNet dataset.

**Claims And Evidence:**

The claims made in the submission are accurate, convincing and clear evidence. The authors take a close look at different versions of ImageNet datasets and take the latest one for a detailed examination. The claims are based on the findings in a concrete dataset and seem reasonable.

**Datasets And Benchmarks:**

N/A

**Extended Submissions:**

N/A

**Limitations:**

Although the authors have suggested that the machine learning community should take a look at the found issues, there is a lack of measures as suggestions for the community. For example, the paper could include some discussion on how some new machine learning methods can be used to address those issues.

**Requested Changes:**

I would recommend the authors to discuss more about the concrete actions the machine learning community can take to address the found issues. For example, whether some new techniques in machine learning can be used to solve those problems.

---

### Review · Reviewer_KdLV · 2024-01-10

**Recommendation:** 3
**Confidence:** 2

**Summary Of Contributions:**

In this paper, the authors highlight a commonly overlooked point about investigating the training data itself. Specifically, the paper takes a multifaceted closer look at ImageNet by looking at its versions over the years, its construction, ethical issues as well as its utility to the broader ML community.

I would like to appreciate the effort put in by the authors in understanding the ImageNet dataset in such great detail as well as giving clear motivations to pursue such studies. With the recent rapid progress in ML techniques, it makes sense to revisit the fundamentals and a closer look at the most common datasets can uncover certain consistent issues as well as possibly unlock future optimizations.

**Strengths:**

- The paper provides a historical timeline of ImageNet over the last 14 years. It provides easy references to each updated dataset while also allowing us to use it as a guide to better understanding differences in model performance.
- Having historical guidance has several benefits. As the authors mention it allows us to trace the various changes to the dataset due to criticisms of the dataset’s accuracy (labelling issues) or the inclusion of offensive or even illegal visual material. Knowing the differences makes it easier to identify or discard datasets for our particular task.
- Having the training dataset section is a good idea. Knowing that different versions of the same dataset have been used to compare methods can help identify finer bias points.
- The authors bring up a crucial point that human images cannot be boiled down to simple categories as often in such a process important context is missing. Another finer point included in the paper is the missing human consent when such images are allowed to be in such a publicly available dataset.
- The broader impact statement includes important discussion points that should be investigated further. Some of my thoughts have been added to the limitations section.

Overall, the paper handles an important topic well and invites a much-needed discussion on dataset terminology, impact and performance.

**Broader Impact Concerns:**

The authors include a thorough broader impact statement and I find it mostly sufficient. Some more comments have been highlighted in the limitations section.

**Claims And Evidence:**

The authors provide thorough references, often adding appropriate context to their findings. I find the paper accurate and clearly organized and written.

**Datasets And Benchmarks:**

The authors do not propose a new dataset but rather discuss their findings on the existing ImageNet dataset and its various versions.

**Extended Submissions:**

To my knowledge, this is a standalone submission.

**Limitations:**

The section below contains several comments regarding the paper. Although, some of them point towards weaknesses in the paper, most of these comments are meant as follow-up questions to the work done and its future extensions.
- I am wondering whether there is a partial benefit to keeping some non-ideal material in a dataset (for instance poorly labelled, mildly offensive content). In real-world scenarios, such data points often crop up and there are research areas for handling such issues. For instance, membership inference is often used to identify people’s participation in datasets while differential privacy is a popular technique to obfuscate people’s identities. These research areas can surely benefit from the people category in ImageNet. Similar, arguments can be made for other non-ideal content. To be clear, I do not think such content should be widely publicized and a possible alternative is to use synthetically generated content or maintain different versions of the dataset. Another option is to require permissions for accessing sensitive datasets by requiring approvals (from say a graduate advisor) such as the one required in clinical study datasets. Could the authors include a section with their thoughts regarding this possibility? I believe there is some discussion in the conclusions about this topic but a more substantiative section will be useful.
- The authors bring up a good point in section 4 where they mention that most methods are not using the most recent ImageNet dataset. Could there be practical reasons for the lack of such evaluations such as lack of reproducible code or simply being able to evaluate methods against other similar methods? Is the solution simply devising an online where evaluations occur on the fly such as the ones used by LLM leaderboards? Could the authors include their thoughts on the possible solutions to the problems mentioned in this study?
- In the training dataset section (4) can the authors include a figure demonstrating the accuracy of dataset correlations (for one popular task)?
- Also, given that several papers use or even highlight their augmented/modified datasets to extract more performance from models, could the authors comment on the importance of such a method regarding ImageNet? To my previous point, the accuracy-dataset graph can highlight some of these benefits.
- Can the authors discuss better methods for handling people-related content? Are there alternative definitions possible instead of simple people categories? I believe there has been some work with LLMs to add a descriptive output per image along with a class. Can such work alleviate major concerns about categorization?
- Minor point: the keywords are missing after the abstract.

**Requested Changes:**

Most of my requested changes/suggestions have been included in the limitations section. In particular, I would prefer to see the inclusion of sections highlighting solutions to the problems associated with the Imagenet dataset and its broader deployment in the updated version. I have added more details in the limitations section.

---

### Review · Reviewer_A6J1 · 2024-01-10

**Recommendation:** 3
**Confidence:** 2

**Summary Of Contributions:**

The paper summarizes how the ImageNet dataset has been updated and curated over the years and the reflection of these changes in the research community. Specifically, the paper provides a comprehensive comparison between each version of the ImageNet dataset, including the unofficial versions created by different researchers, and highlights the important updates. For instance, in the Winter 2021 version of the dataset, over a million images of people were removed due to abusive and/or dehumanizing labels. The paper then shows how many recent research papers use which versions of the dataset. Surprisingly, it turns out that many papers still use the older version of the dataset without the problematic images removed.

**Strengths:**

The paper analyzes one of the most commonly used datasets in the vision domain. It presents a comprehensive survey on how ImageNet dataset has evolved through the years, how certain problems were eliminated over time, and also mentions some remaining problems.

**Broader Impact Concerns:**

The Broader Impact section in the paper is sufficient.

**Claims And Evidence:**

Yes

**Datasets And Benchmarks:**

N/A

**Extended Submissions:**

N/A

**Limitations:**

While it touches on an important problem, dataset curation, and dataset quality, there is no new idea or methodology proposed in the paper.

**Requested Changes:**

My main concern is limited novelty as the paper provides a survey on a dataset, rather than proposing anything new. It could be a great fit for a venue that supports survey papers.